# *"If it's necessary, it has to be done. And that's for the physician to decide, not me."* Imaging techniques in monitoring routines in coronary heart disease and post-stroke patients: A qualitative interview study from the patients' perspective

Laura Rink[1]*, Julia Heisig[2], Ciara Elisa Fink[1], Carolin Gonschorek[2], Veronika van der Wardt[2], Annika Viniol[2], Thomas Kühlein[1], Susann Hueber[1]

1 Institute of General Practice, Friedrich-Alexander-Universität Erlangen-Nürnberg (FAU), Erlangen, Germany, 2 Department of Primary Care, University of Marburg, Marburg, Germany

* Laura.Rink@uk-erlangen.de

## Abstract

### Background

Chronic conditions such as coronary heart disease and stroke impose a significant burden on individuals and healthcare systems. Regular monitoring, including imaging techniques such as echocardiography and carotid duplex sonography, is used in follow-up care to detect disease progression and guide treatment decisions. Although these procedures are thought to provide clinical benefit, evidence-based guidelines provide limited recommendations for their routine use. This raises concerns about potential overdiagnosis, unnecessary healthcare costs and the psychological impact of frequent medical examinations. This study explores how patients with coronary heart disease and post-stroke experience and perceive these monitoring practices, focusing on their expectations, emotional responses and perceived benefits.

### Methods

A qualitative study was conducted using semi-structured interviews with patients who had undergone echocardiography or carotid duplex sonography within the past three years. Participants were recruited in two regions in Germany. Interviews were transcribed and analysed following Braun and Clarke's reflexive thematic analysis approach.

### Results

Fourteen patients were interviewed. Participants generally found monitoring to be beneficial and reassuring. They expressed confidence in their physicians'

**Data availability statement:** The relevant data can be found in the paper and its Supporting information files. Transcripts can be accessed here: 10.5281/zenodo.17415475.

**Funding:** The research presented in the paper was part of the project ChroMo ('Monitoring-Routinen bei Menschen mit chronischen Erkrankungen: Bestandsaufnahme und Fahrplan für die Zukunft' [Monitoring routines for people with chronic diseases: Assessment and roadmap for the future]) and was funded by the Innovation Fund of the Federal Joint Committee ('Gemeinsamer Bundesausschuss': grant 01VSF22046). The funders had no role in study design, data collection and analysis, decision to publish, or preparation of the manuscript.

**Competing interests:** The authors have declared that no competing interests exist.

**Abbreviations:** CCM, Chronic Care Model; CHD, Coronary heart disease; TIA, Transient ischaemic attack; GP, General practitioner; SDM, Shared decision making.

recommendations and felt that the frequency of monitoring was appropriate. Patients left decisions about their regular monitoring entirely to their physicians. A hypothetical reduction raised concerns about the rationing of care. In addition, monitoring was perceived as influencing therapeutic decisions, reinforcing the perceived need for monitoring.

## Conclusions

Patients' positive attitudes towards monitoring are shaped by their perception of medical necessity, which is primarily influenced by their physicians. Given patients' reliance on physician advice, the responsibility for refining monitoring strategies lies primarily with physicians, who should critically assess indications to avoid unnecessary investigations.

## Introduction

Chronic conditions are associated with limitations of functioning [1,2], increased mortality [3] and rising healthcare costs [4]. Over the last decades, healthcare systems have adopted proactive approaches for the management of chronic diseases, such as the Chronic Care Model (CCM) [5]. The CCM emphasises coordinated, patient-centred care to improve long-term outcomes and quality of life [6]. Monitoring plays an essential role in the CCM and in managing patients with chronic conditions in general. It is believed that regular monitoring, including procedures such as diagnostic imaging, laboratory tests and clinical assessments, can anticipate potential deteriorations in the course of chronic diseases. This is based on the underlying assumption that regular monitoring may detect disease progression, prevent potential complications and improve treatment outcomes [7].

Two chronic conditions with particularly high burdens on both individuals and society are coronary heart disease (CHD) and stroke: while stroke is a strong predictor of long-term disability [8] and death [9], CHD remains a major cause of cardiovascular complications [10]. Even if monitoring examinations in such conditions seem plausible, they should not be used uncritically. The relevant guidelines for CHD [11,12] and stroke [13–15] do not provide sufficient and/or evidence-based recommendations for the use of imaging practices like echocardiography and carotid duplex sonography as follow-up routines. However, these monitoring procedures are performed as follow-up care in Germany, leading to concerns about unnecessary treatment consequences, increased healthcare costs, waste of workforce resources, and potential overdiagnosis.

Besides clinical and economic considerations, this raises questions about the value of routine monitoring for patients and its impact on their experiences and perceptions. Studies have shown that patients with chronic conditions often experience increased levels of psychological distress and anxiety [16,17]. This anxiety may be exacerbated by the constant need for medical control examinations, adding to the overall burden of chronic conditions.

The aim of this study was to explore how patients with CHD, post-stroke or transient ischaemic attack (TIA) experience and evaluate their monitoring routines, focusing on the imaging practices of echocardiography and carotid duplex sonography. In Germany, these imaging techniques are carried out by mostly self-employed specialists (cardiologists and neurologists) in outpatient practices. By examining patients' views and the factors that shape their perceptions, this study intends to understand their beliefs about the benefits and potential drawbacks of those monitoring practices, as well as the overall impact of monitoring and its frequencies on their reported health and satisfaction with care.

## Methods

To approach an answer to our research question, we conducted a qualitative study. Semi-structured interviews were conducted to capture the perspectives of patients with CHD and post-stroke on regular echocardiography and carotid duplex sonography monitoring. This method was chosen for its flexibility, allowing participants to express their personal experiences of monitoring routines in detail while allowing the researchers to explore emerging themes and nuances in greater depth. This approach was followed to gain in-depth insights into patients' experiences, attitudes and perceptions, in line with the principles of reflexive thematic analysis by Braun and Clarke [18,19]. Reporting is based on the checklist 'Consolidated criteria for reporting qualitative research' (COREQ) (S1 Checklist) [20].

The interviews were part of a healthcare research project analysing monitoring routines in Germany and developing recommendations (project ChroMo: Monitoring routines for people with chronic diseases: Assessment and roadmap for the future).

The interview study was approved by two local ethics committees (Philipps-Universität Marburg, ref. 23–94 BO and Friedrich-Alexander-Universität Erlangen-Nürnberg, ref. 23-204-Bn). Informed written consent was obtained from all participants.

### Participants and recruitment

Participants included adult patients diagnosed with CHD, stroke or TIA who had undergone echocardiography or carotid duplex sonography monitoring at least three times within the past three years. Diagnoses of CHD, stroke, or TIA were based on patient self-reporting during recruitment. Patients were recruited through general practitioners (GPs) in the regions of Hesse (central, northern and eastern Hesse) and northern Bavaria (Franconia), Germany. Recruitment took place between 2 October 2023 and 31 March 2024. It was facilitated by direct invitations from GPs, as well as posters and flyers in GP practices. In addition, a call for studies was published in a local newspaper. Patients were offered 30 Euros for their participation. Interested persons contacted the study centres in Erlangen or Marburg by phone or email.

There was no prior personal relationship between the interviewers and the participants. Contact was made exclusively during the recruitment process, either via GP practices or at the participants' initiative. The participants were informed in advance about the study's aim and purpose. They were also told that the interviewers were not physicians.

### Data collection

An interview guide with open-ended questions was developed and discussed by a multidisciplinary team (with backgrounds in medicine, psychology, sociology and biology). The interview guide was reviewed and pre-tested with a patient representative (female) who supported the study. She gave feedback on the comprehensibility of the questions, the flow of the interview and the time frame. The interview guide (S2 Text) was designed to explore patients' experiences of monitoring routines, perceived benefits and burdens, and their overall assessment of the monitoring process. In addition, to better understand emotional reactions to the monitoring procedure the guide included a photo elicitation [21]. The pictures showed, for example, a stormy sea, a mountain landscape or a hammock, and were selected by the patient representative and discussed with the research team.

Data were collected through semi-structured interviews conducted by LR and JH (both female, research associates, highly experienced in conducting qualitative interviews), and CF (female, research assistant, trained in conducting qualitative interviews). Interviews were conducted either in participants' homes, at one of the study centres, via online Zoom call, or via telephone, depending on the participant's preference.

Interviews were audio-recorded, subsequently transcribed verbatim, and pseudonymised to ensure confidentiality. Postscripts were prepared after each interview to document observations on the dynamics of the conversation or relevant contextual factors. The transcripts were not sent back to the participants for correction. Participants did not receive any feedback on the preliminary results, but were offered the opportunity to be informed about the results after the end of the study.

### Data analysis

The analysis was conducted using MAXQDA (2020) software. The qualitative data analysis followed the thematic framework by Braun and Clarke [18,19], as it is a flexible analytic approach that allows researchers to identify, analyse, and interpret patterns of meaning across qualitative data. It emphasises researcher reflexivity and the active role of interpretation. This method was particularly suited to our study because it enabled both inductive and deductive exploration of participants' experiences and perceptions of monitoring without being bound to a predefined theoretical framework, by following a systematic process of data immersion, coding and theme development. This approach began with an in-depth familiarisation with the interview transcripts, followed by the generation of initial codes that captured meaningful segments of the data. These codes were then organised into potential themes, which were refined through iterative review and comparison. The process emphasised both data-driven insights and theoretical sensitivity, ensuring that emerging themes authentically reflected participants' experiences.

The analysis involved familiarising with the data, generating initial codes, searching for themes, reviewing themes, defining and naming themes, and writing the final report. The analysis was both inductive and deductive, with initial codes derived from the data and thematic structures influenced by the interview guide. LR, JH and CF independently coded the transcripts, and disagreements were resolved through discussion.

## Results

### Participant characteristics

A total of 29 patients responded and agreed to be interviewed. Of these, 14 interviews were scheduled and conducted, comprising individuals with diverse backgrounds in terms of age, gender, disease duration and time of response. Individuals who had not yet undergone, or had completed only one monitoring examination, were excluded during screening, as the study focused on patients with regular monitoring. No repeat interviews were conducted. Detailed demographic characteristics are shown in Table 1. The interviews took place between November 2023 and March 2024 and lasted on average 32 minutes (range 20–51 minutes).

### Emergent themes

Several codes emerged through thematic analysis (S3 Table), followed by the development of key themes and subthemes (S4 Table). The interviews revealed several recurring topics, indicating thematic saturation. The following themes illustrate key aspects of patients' perspectives and are enriched with anchor quotes to provide deeper insight.

**Theme 1: Perception of a high-quality of care, which is organisationally challenging.** Patients expressed a range of feelings about the healthcare system, describing care itself as supportive but organisational aspects as challenging. While acknowledging that they receive high-quality medical care, they also highlighted structural difficulties such as long waiting times or perceived overuse and underuse of medical services. Despite these challenges, many demonstrated a pragmatic acceptance of the system's limitations: "[...] it's always a bit problematic in the healthcare system to get an

**Table 1. Characteristics of participants.**

| Participants (n = 14) | |
|---|---|
| **Sex** | |
| Men | 12 |
| Women | 2 |
| **Age-range** | |
| ≤60 | 1 |
| 61-70 | 3 |
| 71-80 | 5 |
| ≥81 | 5 |
| **Diagnosis*** | |
| CHD | 7 |
| Stroke | 4 |
| TIA | 3 |

* A number of participants had further diagnoses, beside CHD, stroke or TIA.

appointment with some specialists [...] well, I know. I just have to give six months' notice [that I need a new appointment], that's okay. (P-11)".

*Sub-themes: GPs as guides & cardiologists and neurologists as specialized experts:* GPs were perceived mainly, and in some cases more or less completely, as coordinators of care, playing a central role in guiding patients through their treatment pathways. Patients valued their GP's role in managing referrals and overseeing the overall treatment process: "I think a GP has to go in this direction, in management and the rough stuff. (P-10)"

Patients made a distinction between their GP's role as general coordinator and the perceived high-level medical expertise of specialists such as cardiologists and neurologists. Mainly, these specialists were seen as crucial for dealing with more complex or advanced medical problems. The participants reported that echocardiography or carotid duplex sonography monitoring is carried out exclusively by cardiologists and neurologists in an outpatient clinical setting, not in GP practices. The importance of access to specialist care was therefore frequently mentioned, with patients perceiving a need for expertise beyond primary care: "Just like everywhere else in the profession, you need specialists who can do this. Then it's quicker and more accurate. (P-10)"

**Theme 2: Monitoring takes place, but patient engagement varies.** Some patients were very proactive, asking detailed questions and keeping track of their test results, while others preferred to leave decisions entirely to their physicians: "And if necessary, I also asked about it, so I already knew what it was about. And I also asked what the measurement was, what the value was and what range it was in, depending on the situation. (P-03)" Nevertheless, monitoring takes place regardless of the extent to which patients deal with their illness and whether they question their medical care or not: "Yes, just as I said, if it's necessary, it has to be done. And that's for the physician to decide, not me. (P-02)"

**Theme 3: Positive appraisal of the monitoring frequency.** Patients generally felt that the frequency of their monitoring routines was appropriate and adapted to their individual health status. Frequencies between every three months and every year and a half have been reported. The participants were confident that their physicians were managing their monitoring according to their medical and personal needs: "No, it's enough for me. And if I had the impression that I needed more, then that would certainly not be a problem. (P-14)"

**Theme 4: The perceived benefits of monitoring influence the experience and perception of these examinations.** For many patients, the perceived benefits of routine monitoring outweighed any potential organisational

or time burden of monitoring appointments. Regular follow-up provided reassurance and was seen as an essential part of managing their condition. Procedures such as carotid duplex sonography or echocardiography were often described as reassuring: "Well, there is something reassuring when you hear that the plaque formation has either not progressed any further [...] or if it is found that there is perhaps only 20 percent flow and a stent would be necessary. (P-03)"

*Sub-theme: Monitoring provides safety, while hypothetical discontinuation of monitoring faces concerns:* Many patients saw regular monitoring as a key part of their care, providing safety. When asked during the interview about a hypothetical discontinuation or reduction of monitoring examinations, they reacted with concern. The prospect of stopping or reducing imaging tests such as echocardiography or carotid duplex sonography was associated with worries about health care rationing and uncertainty about their ongoing care: "No, that wouldn't actually be okay with me. I think it's important to have this additional certainty that nothing significant has deteriorated. (P-12)"

**Theme 5: Monitoring examinations may have therapeutic consequences.** Although monitoring was perceived by patients primarily as a diagnostic tool, some patients reported that test results directly influenced medication adjustments or led to referral for further intervention. These therapeutic consequences reinforced the perceived importance of monitoring to patients: "[...] In this respect, I am reassured that the next step was taken as a consequence of this examination in the practice. So, I am also happy that this could be done so quickly. (P-06)"

However, it remained unclear whether the consequences described by patients were directly attributable to echocardiography or carotid duplex ultrasonography, to other interventions or to the sum of interventions during consultations.

## Discussion

This study provides valuable insights into how patients with CHD or post-stroke experience evaluate routine monitoring procedures such as echocardiography and carotid duplex sonography. Generally, patients perceived monitoring as beneficial, providing reassurance about their health status and contributing to a sense of safety. The participants considered the frequency of their monitoring to be appropriate, while any potential reduction or discontinuation of monitoring tests was met with scepticism, often linked to concerns about rationing of care by their physicians.

The general satisfaction of the participating patients in our study with their regular imaging techniques is in line with findings from other studies: Non-invasive imaging procedures are perceived as unproblematic and generally pleasant, contributing to higher acceptance and overall patient satisfaction. Patients benefit from the use of visual reports, leading to a better understanding of their health condition [22].

Moreover, our findings are consistent with previous research showing that patients often overestimate the benefits of medical interventions while underestimating potential harms [13]. This optimistic bias [23] may contribute to patients' perception of safety and reinforces the impression that frequent monitoring is inherently beneficial. Both CHD and post-stroke patients in our study described feeling reassured and safe due to regular monitoring. Compared with research on diagnostic screening and testing, this is consistent with patients having a high level of confidence in screening and medical testing procedures. They were seen as enabling further steps if needed, such as early intervention, and providing on the other hand reassurance if nothing was found [24–26]. This also supports the findings in our research that patients would feel uncertain if existing monitoring procedures were reduced. It can also be assumed that the face-to-face contact with the physician during the examination and the regular consultations resulting from monitoring make a significant contribution to the feeling of safety.

Another finding of our study is that some patients perceived imaging techniques as necessary because they led, or they thought it might lead, to seemingly necessary therapeutic interventions. Studies have shown that physicians also overestimate the benefits of medical interventions [27]. Physicians and patients need to know more about the risks of medical interventions to make shared decisions. An effective communication about realistic outcomes can help reduce misunderstandings [28,29]. By fostering a transparent dialogue, physicians therefore may help patients to develop a better understanding of the necessity and limitations of monitoring and its role as a diagnostic or surveillance measure. Patients may

benefit from being involved in discussions about the need and frequency of monitoring routines, with healthcare providers providing clear explanations of both the expected benefits and potential risks.

Given the emotional importance of monitoring on patients' side, our study supports the need for integration of shared decision making (SDM) into follow-up care. Communicating on ideas, concerns and expectations in counselling and integrating these aspects into SDM could support risk communication in order to make possible disadvantages of monitoring understandable to patients. Findings from research on deprescribing, which focuses on reducing or discontinuing inappropriate or potentially harmful medications, highlight the importance of SDM in adjusting or withdrawing treatments [30,31]. Deprescribing thereby is not about withholding effective therapies from eligible patients. Rather, it is a proactive, patient-centred process that embraces the inherent uncertainties [32]. The results of our study suggest that similar principles could be applied to monitoring routines, particularly with regard to potential reductions when there is no evidence to support the need for them. Further research is needed to assess whether the principles of SDM for deprescribing also apply to diagnostic and monitoring routines. However, as a result of our study, the key issue seems that patients perceive monitoring as essential primarily because it is recommended by their physicians. If monitoring is subsequently reduced or withdrawn, patients may perceive this as a loss rather than an improvement in care. In essence, monitoring provides reassurance for anxiety that is initially created by the very perception of its necessity. Physicians therefore have a crucial role to play in both generating and alleviating this anxiety. This reflects the principle of deprescribing: medications that are never prescribed in the first place do not need to be stopped later. Similarly, unnecessary monitoring that is not initiated does not need to be justified or withdrawn. Consequently, the responsibility for refining monitoring strategies lies primarily with specialists, who must critically assess indications to avoid unnecessary investigations in the first place.

## Strengths and limitations

A key strength of this study is the emphasis on patients' emotions and perceptions, which were central throughout the research process. The use of visual prompts highly supported to facilitate discussion of feelings and experiences. Notably, a patient representative was actively involved in all key stages of the study, contributing personal experiences as a patient into, for example, the interview guide development.

The semi-structured approach of the interviews provided flexibility and allowed for a dynamic and engaging exchange between participants and researchers. In addition, participants were given a choice of interview settings, and participants with health needs were allowed to have a family member present during the interview. Interviews were conducted either in participants' homes, at one of the study centres, via online Zoom call, or via telephone, depending on participant preference and health status. This flexibility ensured accessibility and comfort for all participants.

One limitation of the study is the relatively homogeneous sample. Certain demographic groups may be underrepresented: the predominance of male participants limits the ability to generalise findings across genders. In addition, younger patients were not represented in the sample. Moreover, people who volunteer to participate in health-related research likely tend to have a higher level of health awareness and engagement with their condition and the healthcare system than the general population.

Given the qualitative research design, in which participants were asked to recall past experiences and emotions retrospectively, the possibility of recall bias cannot be excluded. This inherent limitation should be taken into account when interpreting the findings.

## Conclusions

This study highlights the complexity of patients' experiences of routine monitoring in CHD and post-stroke care. While monitoring is widely perceived as beneficial, its role extends beyond its clinical function to include emotional and psychological reassurance. The findings highlight the need for evidence-based monitoring strategies that balance psychological aspects with patient expectations. However, this study shows that patients generally comply with what their physicians

deem necessary. Therefore, the key to optimising monitoring practices lies primarily with specialists. Although patients can be made aware that not all monitoring may be necessary, they are unlikely to act against medical advice. The responsibility for appropriate monitoring therefore lies with specialists, who must ensure that monitoring strategies are evidence-based, and consistent with both clinical needs and patient-centred care.

## Supporting information

**S1 Checklist. Consolidated criteria for reporting qualitative studies (COREQ): 32-item checklist.**
(DOCX)

**S2 Text. Interview guide.**
(DOCX)

**S3 Table. List of codes.**
(DOCX)

**S4 Table. Themes.**
(DOCX)

## Acknowledgments

We thank all patients who participated in the interviews. We are very grateful for the support of our patient representative Astrid Palupski.

## Author contributions

**Conceptualization:** Laura Rink, Julia Heisig, Veronika van der Wardt, Annika Viniol, Susann Hueber.

**Data curation:** Laura Rink, Julia Heisig, Ciara Elisa Fink.

**Formal analysis:** Laura Rink, Julia Heisig, Ciara Elisa Fink, Carolin Gonschorek.

**Funding acquisition:** Veronika van der Wardt, Annika Viniol, Thomas Kühlein, Susann Hueber.

**Investigation:** Laura Rink, Julia Heisig, Ciara Elisa Fink.

**Methodology:** Laura Rink, Julia Heisig, Veronika van der Wardt, Annika Viniol.

**Project administration:** Laura Rink, Julia Heisig.

**Resources:** Veronika van der Wardt, Annika Viniol, Thomas Kühlein, Susann Hueber.

**Supervision:** Veronika van der Wardt, Annika Viniol, Thomas Kühlein, Susann Hueber.

**Validation:** Laura Rink, Julia Heisig, Ciara Elisa Fink, Carolin Gonschorek.

**Visualization:** Laura Rink.

**Writing – original draft:** Laura Rink.

**Writing – review & editing:** Julia Heisig, Ciara Elisa Fink, Carolin Gonschorek, Veronika van der Wardt, Annika Viniol, Thomas Kühlein, Susann Hueber.

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
