## [Decision Letter · Decision Letter 0]

3 Sep 2025

Dear Dr. Rink,

Thank you for submitting your manuscript to PLOS ONE. After careful consideration, we feel that it has merit but does not fully meet PLOS ONE’s publication criteria as it currently stands. Therefore, we invite you to submit a revised version of the manuscript that addresses the points raised during the review process.

**Please note that we have only been able to secure a single reviewer to assess your manuscript. We are issuing a decision on your manuscript at this point to prevent further delays in the evaluation of your manuscript. Please be aware that the editor who handles your revised manuscript might find it necessary to invite additional reviewers to assess this work once the revised manuscript is submitted. However, we will aim to proceed on the basis of this single review if possible. **
**Please carefully revise your manuscript to address all of the reviewer comments below, including providing additional details of the interview location in your methods, and the strengths and limitations of your study in the discussion. **

We look forward to receiving your revised manuscript.

Kind regards,

Jennifer Tucker, PhD

Staff Editor

PLOS ONE

**Journal Requirements:**

1. When submitting your revision, we need you to address these additional requirements. Please ensure that your manuscript meets PLOS ONE's style requirements, including those for file naming. The PLOS ONE style templates can be found at https://journals.plos.org/plosone/s/file?id=wjVg/PLOSOne_formatting_sample_main_body.pdf and https://journals.plos.org/plosone/s/file?id=ba62/PLOSOne_formatting_sample_title_authors_affiliations.pdf 2. Thank you for stating the following in the Acknowledgments Section of your manuscript: We thank all patients who participated in the interviews. We are very grateful for the support of our patient representative Astrid Palupski. We acknowledge financial support by Deutsche Forschungsgemeinschaft and Friedrich-Alexander-Universität Erlangen-Nürnberg within the funding programme “Open Access Publication Funding”. We note that you have provided funding information that is not currently declared in your Funding Statement. However, funding information should not appear in the Acknowledgments section or other areas of your manuscript. We will only publish funding information present in the Funding Statement section of the online submission form. Please remove any funding-related text from the manuscript and let us know how you would like to update your Funding Statement. Currently, your Funding Statement reads as follows: The research presented in the paper was part of the project ChroMo (‘Monitoring-Routinen bei Menschen mit chronischen Erkrankungen: Bestandsaufnahme und Fahrplan für die Zukunft’ [Monitoring routines for people with chronic diseases: Assessment and roadmap for the future]) and was funded by the Innovation Fund of the Federal Joint Committee (‘Gemeinsamer Bundesausschuss’: grant 01VSF22046). The funders had no role in study design, data collection and analysis, decision to publish, or preparation of the manuscript. Please include your amended statements within your cover letter; we will change the online submission form on your behalf. 3. When completing the data availability statement of the submission form, you indicated that you will make your data available on acceptance. We strongly recommend all authors decide on a data sharing plan before acceptance, as the process can be lengthy and hold up publication timelines. Please note that, though access restrictions are acceptable now, your entire data will need to be made freely accessible if your manuscript is accepted for publication. This policy applies to all data except where public deposition would breach compliance with the protocol approved by your research ethics board. If you are unable to adhere to our open data policy, please kindly revise your statement to explain your reasoning and we will seek the editor's input on an exemption. Please be assured that, once you have provided your new statement, the assessment of your exemption will not hold up the peer review process. 4. Please include captions for your Supporting Information files at the end of your manuscript, and update any in-text citations to match accordingly. Please see our Supporting Information guidelines for more information: http://journals.plos.org/plosone/s/supporting-information. 5. If the reviewer comments include a recommendation to cite specific previously published works, please review and evaluate these publications to determine whether they are relevant and should be cited. There is no requirement to cite these works unless the editor has indicated otherwise. 

Reviewers' comments:

**Comments to the Author**

1. Is the manuscript technically sound, and do the data support the conclusions?

Reviewer #1: Yes

2. Has the statistical analysis been performed appropriately and rigorously?

Reviewer #1: N/A

3. Have the authors made all data underlying the findings in their manuscript fully available?

Reviewer #1: Yes

4. Is the manuscript presented in an intelligible fashion and written in standard English?

Reviewer #1: Yes

**Reviewer #1:**  Introduction - please include a sentence or two about how CHD and stroke patients share characteristics. Please explain that CHD can be a risk factor for stroke, so interviewing both cohorts could help with longitudinal monitoring to predict changes before stroke, in the case of patients with CHD only. Please include information about the differences between stroke and TIA, since TIA patients were included in this analysis. TIA is also a risk factor for stroke, so it should be considered the way CHD is - a potential diagnosis to monitor to prevent a stroke.

Did your patients with TIA also participate in regular echocardiography and carotid duplex sonography monitoring? What does "regular monitoring" intervals mean, i.e. how frequent? Were these procedures performed in a clinical setting or at home?

Methods - please provide a paragraph explaining the rationale for why reflexive thematic analysis by Braun and Clarke is best suited for this project. Define this analysis approach for the study.

Participants and recruitment - how were diagnoses defined? Did you obtain diagnosis from electronic medical records or Patient self-reporting?

Data analysis - Could you start this section with the sentence so readers know which tool was used: "The analysis was conducted using MAXQDA (2020) software." ?

Results - could you create a participant enrollment diagram that shows how many patients you contacted, the 29 who replied (and how they dropped out), the 14 who completed the study, and then 2 separate boxes showing the number of study completion participants by diagnosis (CHD diagnosis vs stroke diagnosis)?

Could you give the average/mean for this: "lasted between 20 and 51 minutes."?

Is there any information about participant education levels? Do you have ethnicity information?

Line 222 is missing a leading " for the quote.

Discussion - after line 281, you mention "The results of our study suggest that similar principles could be applied to monitoring routines, particularly". It would be beneficial to include a few sentences about the state of monitoring routines in real-world settings, perhaps by citing a review involving ischemic heart disease and CHD as well as a novel TIA/stroke mood monitoring study (below). The range of remote monitoring digital tools should be discussed here.

#1 - Indraratna P, Tardo D, Yu J, Delbaere K, Brodie M, Lovell N, Ooi SY

Mobile Phone Technologies in the Management of Ischemic Heart Disease, Heart Failure, and Hypertension: Systematic Review and Meta-Analysis

JMIR Mhealth Uhealth 2020;8(7):e16695

doi: 10.2196/16695

#2 - Zawada S, Acosta J, Collins C, Dumitrascu O, Harahsheh E, Hagen C, et al. Real-

World Smartphone Data Predicts Mood After Ischemic Stroke and Transient Ischemic Attack

Symptoms and May Constitute Digital Endpoints: A Proof-of-Concept Study. Mayo Clinic

598 Proceedings: Digital Health. 2025;3(3)

Strengths and limitations - were all interviews conducted in patients' homes?

**Do you want your identity to be public for this peer review?** For information about this choice, including consent withdrawal, please see our Privacy Policy

Reviewer #1: No

---

## [Author Response · Author response to Decision Letter 1]

22 Oct 2025

Reviewer #1:

Introduction - please include a sentence or two about how CHD and stroke patients share characteristics. Please explain that CHD can be a risk factor for stroke, so interviewing both cohorts could help with longitudinal monitoring to predict changes before stroke, in the case of patients with CHD only. Please include information about the differences between stroke and TIA, since TIA patients were included in this analysis. TIA is also a risk factor for stroke, so it should be considered the way CHD is - a potential diagnosis to monitor to prevent a stroke.

Response: Thank you for your thoughts on this. Both conditions have been chosen as examples for monitoring routines. In this context, it was considered whether the conditions are relevant to primary care and whether the monitoring routines can be reflected in billing data for another part of the project. From our perspective, there was no thematic connection between the two conditions. Therefore, although you were absolutely right to describe these connections, we would not include them in the introduction, as they were not part of the study question and methodology.

Did your patients with TIA also participate in regular echocardiography and carotid duplex sonography monitoring? What does "regular monitoring" intervals mean, i.e. how frequent? Were these procedures performed in a clinical setting or at home?

Response: Yes, all patients included in our study had undergone at least three echocardiography or carotid duplex sonography examinations within the last three years. That was one of the recruitment criteria. Monitoring intervals ranged between every three months and every eighteen months (see RESULTS – Theme 3). All examinations were performed in outpatient clinical settings by cardiologists or neurologists.

Changes in manuscript:

• METHODS “Participants included adult patients diagnosed with CHD, stroke or TIA who had undergone echocardiography or carotid duplex sonography monitoring at least three times within the past three years.”

• RESULTS “The participants reported that echocardiography or carotid duplex sonography monitoring is carried out exclusively by cardiologists and neurologists in an outpatient clinical setting, not in GP practices.”

Methods - please provide a paragraph explaining the rationale for why reflexive thematic analysis by Braun and Clarke is best suited for this project. Define this analysis approach for the study.

Response: A paragraph defining and justifying the use of reflexive thematic analysis has been added to the Data Analysis section. This addition clarifies the flexibility and reflexivity of this method and its alignment with our research aims.

Changes in manuscript: “The qualitative data analysis followed the thematic framework by Braun and Clarke (18, 19), as it is a flexible analytic approach that allows researchers to identify, analyse, and interpret patterns of meaning across qualitative data. It emphasises researcher reflexivity and the active role of interpretation. This method was particularly suited to our study because it enabled both inductive and deductive exploration of participants’ experiences and perceptions of monitoring without being bound to a predefined theoretical framework, by following a systematic process of data immersion, coding and theme development.”

Participants and recruitment - how were diagnoses defined? Did you obtain diagnosis from electronic medical records or Patient self-reporting?

Response: Diagnoses were based on patient self-reporting during recruitment. As the study focused on patients’ experiences rather than clinical validation, self-reported diagnoses of CHD, stroke, or TIA were considered appropriate.

Changes in manuscript: “Diagnoses of CHD, stroke, or TIA were based on patient self-reporting during recruitment.”

Data analysis - Could you start this section with the sentence so readers know which tool was used: "The analysis was conducted using MAXQDA (2020) software."?

Response: We agree and have moved this sentence to the beginning of the data analysis section.

Results - could you create a participant enrollment diagram that shows how many patients you contacted, the 29 who replied (and how they dropped out), the 14 who completed the study, and then 2 separate boxes showing the number of study completion participants by diagnosis (CHD diagnosis vs stroke diagnosis)?

Response: We appreciate this suggestion. However, it was not possible to determine the total number of individuals reached by our recruitment efforts, as participants were recruited a public press calls and several general practices. Therefore, we could not create an enrollment flowchart.

We clarified in the Results section how participants were selected and why some were not included, describing the screening criteria and selection process in more detail.

Changes in manuscript: “Of these, 14 interviews were scheduled and conducted, comprising individuals with diverse backgrounds in terms of age, gender, disease duration and time of response. Individuals who had not yet undergone, or had completed only one monitoring examination, were excluded during screening, as the study focused on patients with regular monitoring”.

Could you give the average/mean for this: "lasted between 20 and 51 minutes."?

Response: Thank you for the suggestion. We added the mean interview duration to the Results section.

Changes in manuscript: “The interviews took place between November 2023 and March 2024 and lasted on average 32 minutes (range 20-51 minutes).”

Is there any information about participant education levels? Do you have ethnicity information?

Response: As the study focused on patients’ experiences rather than sociodemographic differences, education and ethnicity data were not collected.

Line 222 is missing a leading " for the quote.

Response: Thank you for catching this. The missing quotation mark in line 222 has been added.

Discussion - after line 281, you mention "The results of our study suggest that similar principles could be applied to monitoring routines, particularly". It would be beneficial to include a few sentences about the state of monitoring routines in real-world settings, perhaps by citing a review involving ischemic heart disease and CHD as well as a novel TIA/stroke mood monitoring study (below). The range of remote monitoring digital tools should be discussed here.

#1 - Indraratna P, Tardo D, Yu J, Delbaere K, Brodie M, Lovell N, Ooi SY

Mobile Phone Technologies in the Management of Ischemic Heart Disease, Heart Failure, and Hypertension: Systematic Review and Meta-Analysis JMIR Mhealth Uhealth 2020;8(7):e16695 doi: 10.2196/16695

#2 - Zawada S, Acosta J, Collins C, Dumitrascu O, Harahsheh E, Hagen C, et al. Real-World Smartphone Data Predicts Mood After Ischemic Stroke and Transient Ischemic Attack Symptoms and May Constitute Digital Endpoints: A Proof-of-Concept Study. Mayo Clinic 598 Proceedings: Digital Health. 2025;3(3)

Response: We thank you for this thoughtful suggestion. However, our study specifically focused on physician-led, imaging-based monitoring in outpatient care. The integration of remote monitoring technologies lies outside the scope of our research. To maintain conceptual and methodological focus, we therefore decided not to expand the discussion in this direction.

Strengths and limitations - were all interviews conducted in patients' homes?

Response: We clarified the settings of the interviews in the strengths and limitations section.

Changes in manuscript: “Interviews were conducted either in participants’ homes, at one of the study centres, via online Zoom call, or via telephone, depending on participant preference and health status. This flexibility ensured accessibility and comfort for all participants.”

---

## [Decision Letter · Decision Letter 1]

23 Nov 2025

"If it's necessary, it has to be done. And that's for the physician to decide, not me."

Imaging techniques in monitoring routines in coronary heart disease and post-stroke patients:  A qualitative interview study from the patients' perspective

PONE-D-25-31747R1

Dear Dr. Rink,

We’re pleased to inform you that your manuscript has been judged scientifically suitable for publication and will be formally accepted for publication once it meets all outstanding technical requirements.

Kind regards,

Redoy Ranjan, MBBS, MRCSEd, Ch.M., MS (CV&TS), FACS

Academic Editor

PLOS ONE

Additional Editor Comments (optional):

Reviewers' comments:

Reviewer's Responses to Questions

**Comments to the Author**

Reviewer #1: All comments have been addressed

2. Is the manuscript technically sound, and do the data support the conclusions?

Reviewer #1: Yes

3. Has the statistical analysis been performed appropriately and rigorously?

Reviewer #1: N/A

4. Have the authors made all data underlying the findings in their manuscript fully available?

Reviewer #1: Yes

5. Is the manuscript presented in an intelligible fashion and written in standard English?

Reviewer #1: Yes

Reviewer #1: Author reply satisfactory. Recommendation to accept paper. Research revisions meet each request outlined in review round 1

**Do you want your identity to be public for this peer review?** For information about this choice, including consent withdrawal, please see our Privacy Policy

Reviewer #1: No

---

## [Editor Report · Acceptance letter]

PONE-D-25-31747R1

PLOS ONE

Dear Dr. Rink,

I'm pleased to inform you that your manuscript has been deemed suitable for publication in PLOS ONE. Congratulations! Your manuscript is now being handed over to our production team.

Kind regards,

on behalf of

Dr. Redoy Ranjan

Academic Editor

PLOS ONE